# The Effect of Cognitive Resource Competition Due to Dual-Tasking on the Irregularity and Control of Postural Movement Components

**DOI:** 10.3390/e21010070

**Published:** 2019-01-15

**Authors:** Thomas Haid, Peter Federolf

**Affiliations:** Department of Sport Science, University of Innsbruck, 6020 Innsbruck, Austria

**Keywords:** postural control, dual-task, working memory, sample entropy, principal component analysis, minimal intervention principle, age effects, tandem stance, neuromuscular control

## Abstract

Postural control research suggests a non-linear, n-shaped relationship between dual-tasking and postural stability. Nevertheless, the extent of this relationship remains unclear. Since kinematic principal component analysis has offered novel approaches to study the control of movement components (PM) and n-shapes have been found in measures of sway irregularity, we hypothesized (H1) that the irregularity of PMs and their respective control, and the control tightness will display the n-shape. Furthermore, according to the minimal intervention principle (H2) different PMs should be affected differently. Finally, (H3) we expected stronger dual-tasking effects in the older population, due to limited cognitive resources. We measured the kinematics of forty-one healthy volunteers (23 aged 26 ± 3; 18 aged 59 ± 4) performing 80 s tandem stances in five conditions (single-task and auditory n-back task; n = 1–4), and computed sample entropies on PM time-series and two novel measures of control tightness. In the PM most critical for stability, the control tightness decreased steadily, and in contrast to H3, decreased further for the younger group. Nevertheless, we found n-shapes in most variables with differing magnitudes, supporting H1 and H2. These results suggest that the control tightness might deteriorate steadily with increased cognitive load in critical movements despite the otherwise eminent n-shaped relationship.

## 1. Introduction

Postural control with the aim of maintaining balance is a complex process in the sense that it depends on available visual, proprioceptive, and vestibular sensory information, and on the adequate neuromuscular response to counteract perturbations [1,2,3,4]. Empirical evidence suggests that cognitive resources are needed for the selection of conflicting sensory information [4,5,6] and for the compensation of impairments or perturbations of the postural control system [7]. It has also been found that performing a dual-task (DT) might interfere with this complex process and lead to increases in postural sway area, velocity and frequency, which are typically interpreted as a deterioration in postural stability [8,9,10,11]. 

However, other research has found postural stability not to suffer from dual-tasking. On the contrary, postural sway was observed to be reduced when performing easy dual tasks [12,13,14,15]. These controversial results are not surprising, since maintaining postural stability must often be performed without extensive attentional focus to the balancing task, as daily situations routinely require the attentional focus for another, simultaneous task. Nevertheless, further challenging participants with more difficult DTs had no beneficial effects or even increased postural sway, resulting in a u-shaped relationship between postural sway and the cognitive demand of the DT [15,16]. It was proposed that this relationship could be the result of the competition for limited cognitive resources; or in other words, an n-shaped relationship between cognitive demand and the efficacy of the postural control system. In detail, it was postulated that the beneficial effects of easy dual-tasking might suggest a more automatized and therefore more effective control [15,17,18,19]. Nevertheless, even an effective and rather automatized control system would still have to rely on some form of cognitive processing [1,5,6,7,15,16], and hence, a difficult DT could lead to cognitive resource competition between the attentional demand of the DT and the cognitive requirements of postural control. However, how and to which extent dual-tasking affects postural control is still subject to debate. 

We believe that in order to study the effects of cognitive DT on postural control the following two aspects are important to be considered: First, the dual-task difficulty levels used in previous studies relied on several different forms of cognitive demands such as listening, memory, reaction time, spatial distinction or calculation [15,16,20,21,22,23,24]. Since previous results suggest that different forms of cognitive tasks display contrasting levels of interaction with postural control [10], this diversity in cognitive resource requirement makes comparisons among studies and even between trials in the same study difficult. Second, center of pressure COP-based variables are very common in postural control research. A great advantage of such analyses is that several interesting aspects of postural control can be quantified based on the COP time-series, since it contains the combined information about body positioning and acting forces [25]. Nevertheless, literature using COP variables is not consistent. For example, on the one hand COP-irregularity has been observed to increase with task difficulty, i.e., higher entropy was found when balancing with eyes closed, on foam, or when dual-tasking [26,27,28]. In addition, COP-entropy was also found to be linked to better health or training status, e.g., young vs. old, healthy vs. concussed or trained vs. non-trained subjects [20,21,22,29,30]. In these studies, higher COP-irregularity was associated with a more adaptable and alert system. On the other hand, however, higher COP-irregularity has also been associated with balancing in lower task difficulty (e.g., eyes open condition) or elderly fallers [20,21,24,31,32]. Here, higher entropy was linked to disordered and less effective postural control. Hence, although COP-variables have proven to be very effective distinguishing groups, their implications on the control of movements are not straight forward, explaining why the literature is not consistent. 

These inconsistencies can be partially explained by findings which suggest that the commonly used COP-irregularity is not a measure that describes only one specific aspect of postural control but can be linked to at least two different aspects of postural control, specifically to the irregularity of the control of certain movement strategies or the mechanical complexity of a movement (the number of different movement strategies that are coordinated [33]). Furthermore, COP-variables reduce the vast amount of degrees of freedom (DOF) into one two-dimensional variable, which does not provide information about the exact segment positioning or movement strategies [34]. This is problematic since a well-established principle for human motor control, the minimal intervention principle [35], suggests that the control system distinguishes between DOF that are task-relevant and DOF that are not task-relevant. Analyzing postural control in variables that characterize actual segment movement patterns, or movement strategies, may therefore yield more conclusive results than COP analyses that condense the many DOF into a two-dimensional time-series. 

To identify relevant movement strategies, research has successfully applied principal component analysis (PCA) to kinematic data with the aim of identifying relevant movement components, called principal movements (PM*_k_*) [36,37,38,39]. It has also been shown, that the respective time-series of these PM*_k_*, specifically principal positions PP*_k_* and principal accelerations PA*_k_*, can be used to compute COP time-series [34], hence they contain the information of COP-trajectories. Furthermore, it could be shown that the COP-irregularity correlates with the irregularity of specific PP*_k_*, showing that to some extent the dynamics of PP are preserved in the COP [33]. Hence, these results also suggest that PCA-based variables can be used to expand the concept of COP-irregularity and might be better suitable to explain aspects of motor control that require considering the minimal intervention principle. Furthermore, PCA-related variables computed on the PA time-series were sensitive enough to identify effects in aging [40] and leg dominance [41], by quantifying how often the control system intervenes (N) and how variable the timing of these interventions is (σ). Again, not the entire neuromuscular control system displayed the effects, but only the control of specific movement components, emphasizing the need to consider the minimal intervention principle. 

In summary, previous findings have linked reduced postural sway to more efficient postural control. Hence, the u-shaped relationship in terms of postural sway suggests an n-shaped relationship in terms of the efficiency of postural control. Furthermore, if high movement irregularity is indeed a sign of a more alert and efficient system as various findings suggest [20,21,22,29,30], the sample entropies of the PP*_k_* and the PA*_k_* (${\mathrm{SaEn}}_{k}^{PP},\;{\mathrm{SaEn}}_{k}^{PA}$) should also display this n-shaped relationship. In addition, tighter movement control has been interpreted as more efficient postural control, i.e., a high number of interventions of the control system (N) and a lower timing variability (σ) were observed in young versus old [40] or in the dominant versus the non-dominant leg [41]. Hence, the N*_k_* should also display the n-shaped relationship, whereas σ*_k_* should display a u-shaped relationship. 

The current study investigates the effect of cognitive dual-tasking on the postural control of a tandem stance for comparable cognitive tasks of increasing difficulty level. Given that previous studies attribute dual-tasking effects in COP time-series to changes in the neuromuscular control and the resulting postural sway, we hypothesized that the n-shaped relationships between the efficacy of postural control and dual-task difficulty should be evident in the all four types of variables (${\mathrm{SaEn}}^{PP},\;{\mathrm{SaEn}}^{PA}$, N and σ) (H1). Based on the minimal intervention principle, we hypothesized that dual-tasking effects will not affect the system as a whole but emerge in specific movement components that are task-relevant (H2), i.e., more critical for maintaining postural stability. Furthermore, in agreement with previous studies, we expected that the postural control system of the older population will display stronger dual-tasking effects in the critical movement components due to their more limited cognitive resources (H3). 

## 2. Materials and Methods

### 2.1. Participants and Measurement Procedures

The analyzed data was a subset of a dataset of 106 participants of a previous, unpublished study that recorded the 3D kinematics of several balance and walking measurements during two visits. The study was conducted in agreement with the Declaration of Helsinki. In particular, the original study design had been approved by an institutionalized ethics review board and prior to any measurements informed written consent was obtained from all participants. Furthermore, the inclusion criteria of the original study were: (i) self-reported good health and neither cardiovascular nor neurophysiological problems; (ii) the participant’s occupation required them to be standing or walking roughly half of their time; (iii) the participant’s age was either in the range of 20–35 years or in the range of 55–70 years; (iv) no regular training for a specific sport (only occasional recreational sports activities). The participants of the selected subset had to additionally have (v) completed all 5 tandem stance trials of one of two visits successfully (no step or arm movements required to maintain balance). A total of 41 participants fulfilled the fifth criterion—23 of the younger age group aged 26 ± 3 years, height 1.7 ± 0.1 meters and weight 68 ± 10 kg, and 18 of the older age group aged 59 ± 4 years, height 1.6 ± 0.1 meters and weight 68 ± 11 kg (mean ± standard deviation). 

Before the measurements, the participants were asked to stand barefoot and to test both tandem stance positions (left foot in front and right foot in front, and the rear foot barely touching the heel of the front foot) to find out which one felt more comfortable. Then, they were instructed to stand in the more comfortable tandem stance position for 80 s and to “stand as still as possible”. In each session they completed five tandem stance trials: one single task condition (ST) and four in different cognitive dual-task conditions (DT*_n_*) of increasing difficulty. The DT*_n_* were auditory n-back working memory tasks with n = 1–4, i.e., via headphones the participants listened to a sequence of 26 seemingly random numbers ranged between 1 and 9, and had to reply with “yes” when the current number was equal to the *n*th digit before. Furthermore, each sequence contained 6 n-back stimuli and the inter-stimulus interval time in the sequence was randomized to avoid subjects falling into a rhythmical routine. Full body kinematics were recorded at 240 frames per second by 8 synchronized video cameras (Motion Analysis Corp., Santa Rosa, CA, USA) using 37 reflective markers (standard 39 plug-in gait marker set without markers on the hands). 

### 2.2. Data Analysis

All data analysis steps were conducted via Matlab R2015a (The MathWorks Inc., Natick, MA, USA). The statistics were conducted using Matlab and IBM SPSS Version 24 (SPSS Inc., Chicago, IL, USA).

#### 2.2.1. Pre-Processing

To fill gaps in the data, a PCA-based reconstruction technique was used [42,43]. Then, both the first and the last 10 s were omitted to reduce possible starting difficulties [44] or teleoanticipation effects due to the end of the trial [45]. Furthermore, the data was down-sampled to 120 Hz to reduce noise amplification due to differentiation. In addition, the trials of subjects standing with their left foot in front were mirrored and relabeled. This ensured that all data sets described trials with the same foot in front. Therefore, unsymmetrically placed markers had to be omitted, resulting in 28 markers (28 markers × 3 dimensions=84 columns) entering the analysis. Anthropometric differences were addressed by subtracting the mean posture and normalizing the data to the participant’s height [33,40]. Finally, each marker was weighed with its relative segment mass [46,47], before concatenating all trials into one input matrix for a kinematic principal component analysis PCA.

#### 2.2.2. Kinematic Principal Component Analysis

Each row of the concatenated data-matrix represents one participant’s normalized posture at a given time-point [37,38,46]. Each column can be interpreted as one coordinate in the high-dimensional posture space. Conducting a PCA determines a new orthonormal basis of this posture space consisting of eigenvectors PC*_k_* that are oriented in the direction of the largest variance and define postural movement components or “principal movements” PM*_k_* [34]. Furthermore, the PCA yields trial-specific scores PP*_k_*(*t*) that represent the positions in posture space with respect to the PC*_k_* and eigenvalues EV*_k_* that quantify the overall contribution of a PM*_k_* to the overall variance. In a similar fashion to standard biomechanics the PP*_k_* can be used to compute principal velocities PV*_k_*(*t*) and principal accelerations PA*_k_*(*t*) [34].

#### 2.2.3. Measures of Postural Control

The PP*_k_* and the PA*_k_* contain the information about posture and postural accelerations over time. It has been shown that they can be used not only to compute the center of pressure COP [34], but to quantify similar aspects of postural control as COP-based irregularity, with the advantage of preserving the information about involved body segment movements [33]. Similarly, in the current study the sample entropy SaEn was computed on both the detrended PP*_k_* and the PA*_k_* of all volunteers (${\mathrm{SaEn}}_{k}^{PP}$ and ${\mathrm{SaEn}}_{k}^{PA}$), as measure of the irregularity of the movement strategies and the irregularity of the neuromuscular control of specific movement components, respectively. The PP*_k_* were detrended by subtracting the floating average taken over 501 data points (around 4 s). The floating averages of the first 250 and last 250 data points were considered equal to the nearest available value. The sample entropy calculation parameters were set to typical values: embedding dimension *m* = 2 and r = 0.2·STD [48] (with STD being the standard deviation of the time-series). The time-lag was set to $\tau =12≙\frac{12}{120}s=100\;\mathrm{ms}$, which is a meaningful timeframe from a physiological point of view [49].

Moreover, we computed two variables on the PA*_k_* that further characterize the neuromuscular control, namely, the number of PA zero crossings N*_k_* and the standard deviation of the time between zero-crossings σ*_k_*. These variables serve as a measure for the tightness of the neuromuscular control; specifically, a “tight” control is considered to have a high number of interventions N and a low timing variability of the interventions. Previous studies using these measures were able to identify age differences in ST tandem stances [40], as well as differences in the control of one-legged static balance [39,41] when comparing dominant to non-dominant leg.

### 2.3. Validity Considerations and Cross-Validation

On average each of the 84 columns of the input matrix contributes 1.2% (1/84) to the overall information. Therefore, only PMs that contributed more than 1.2% were taken into consideration. Furthermore, the vulnerability of these PMs to changes in the input data was evaluated by conducting a leave-one-out cross-validation [50,51,52]. 

A Fourier analysis on the PP*_k_*(*t*) was conducted, revealing that the highest power resided in frequencies up to 3 Hz. However, visible power was still found up to 7 Hz. The PP*_k_*(*t*) and the PA*_k_*(*t*) were therefore filtered with a 6th-order Butterworth filter using a cut-off frequency of 7 Hz before computing the variables. Unfortunately, SaEn*_k_*, N*_k_* and σ*_k_* are all susceptible to noise. However, previous studies show [33,40,41] that while the absolute values of these variables depend highly on the filtering frequency and the chosen entropy parameters, resultant statistical effects do not. 

### 2.4. Statistics

A split plot repeated measures ANOVA was conducted on all variables separately (${\mathrm{SaEn}}_{k}^{PP},$
${\mathrm{SaEn}}_{k}^{PA},$ N*_k_* and σ*_k_*). Normality of the data was tested with both Kolmogorov–Smirnov and Shapiro–Wilk tests. The equality of variances was assessed computing Levene’s tests. If the sphericity criterion was not met (Mauchly’s test) a Greenhouse–Geisser or a Huynh–Feldt correction was performed for epsilon values smaller than 0.75, or greater or equal than 0.75, respectively. We report the *p*-value, partial eta squared $\eta_{p}^{2}$, and the observed power π for all main dual-tasking effects. To simplify the main table, the corrected degrees of freedom and the respective F statistics were not included. For each statistically significant result (p<0.05) a post-hoc analysis was conducted using a Sidak correction. For these we report the increases in percent and the *p*-values.

## 3. Results

### 3.1. PCA Results

The first six PMs contributed over 1.2% to the overall variance each and over 95% cumulatively. All of them were robust, i.e., the orientation of the PC-vectors changed less than 15° in the leave-one-out cross-validation. The movement strategies that these PMs resemble are described in Table 1. In addition, Figure 1, and the two Appendix A “Visualization_PM1-PM3.gif” and “Visualization_PM4-PM6.gif” show graphical representations of these PMs.

### 3.2. Dual-Tasking Effects (n-back)

The DT statistics are summarized in Table 2. Significant results are highlighted with bold font. Following, the post-hoc analysis of the statistically significant dual-tasking effects a presented and visualized in Figure 2. The descriptive statistics of all variables and conditions (mean and standard error) can be found in the Appendix A (“DualTasking_effects.png”). 

The post-hoc analysis of the statistically significant results of Table 2 revealed significant differences between 0-back and 2-back conditions in the sample entropies of the PP_5_ and PP_6_ (${\mathrm{SaEn}}_{5}^{PP}$: +12% (ST to DT_2_), *p* = 0.004; and ${\mathrm{SaEn}}_{6}^{PP}$: +11% (ST to DT_2_), *p* = 0.039). Furthermore, significant effects were found between 2-back and 4-back DT in the sample entropy of PP_6_ (${\mathrm{SaEn}}_{6}^{PP}$: −9% (DT_2_ to DT_4_), *p* = 0.012) and the first two PAs (${\mathrm{SaEn}}_{1}^{PA}$: −3% (DT_2_ to DT_4_), *p* = 0.023; and ${\mathrm{SaEn}}_{2}^{PA}$: −2% (DT_2_ to DT_4_), *p* = 0.020). The variables N_5_ and σ_2_ displayed differences between two DT-condition each (N_5_: −4% (DT_1_ to DT_2_), *p* = 0.026 and σ_2_: −3% (ST to DT_4_), *p* = 0.021). No statistically significant effects were found in the post-hoc analysis of N_6_.

### 3.3. Interaction Effects

Significant DT-age-interaction effects were found in the irregularity of two variables (${\mathrm{SaEn}}_{6}^{PP}$: F_(4,156)_ = 3.14, *p* = 0.016, η_*p*_^2^ = 0.08, π = 0.79; and ${\mathrm{SaEn}}_{5}^{PA}$: F_(3.45,134.00)_ = 3.01, *p* = 0.026, η_*p*_^2^ = 0.07, π = 0.74). In both cases only the older participants displayed significant dual-tasking effects and the expected n-shaped relationship (Figure 3), while the younger age group showed rather constant irregularity values throughout varying dual-task conditions. Specifically, the interaction effects in ${\mathrm{SaEn}}_{6}^{PP}$ showed that the main dual-tasking effects in the irregularity of PP_6_ originated from the older participants. 

### 3.4. Age Effects between Subjects

Age effects were found in one variable in the mediolateral ankle sway (N_2_: F_(1,39)_ = 8.74, *p* = 0.005, η_p_^2^ = 0.18, π = 0.82). In detail, the older participants displayed decreased activity of the control system in three conditions (ST: −12% (old vs. young): *p* = 0.002; DT_1_: −12% (old vs. young): *p* = 0.003; and DT_2_: −9% (old vs. young): *p* = 0.025). Furthermore, significant age effects were found in the movement component that resembles an upper body retraction (${\mathrm{SaEn}}_{3}^{PP}$: F_(1,39)_ = 5.93, *p* = 0.020, η_p_^2^ = 0.132, π = 0.66), originating from the higher sway irregularity of the older group in the two easiest dual-tasking conditions (DT_1_: +18% (old vs. young): *p* = 0.009 and DT_2_: +19% (old vs. young): *p* = 0.002). In addition, age effects were found in the hip/knee strategy (${\mathrm{SaEn}}_{6}^{PP}$: F_(1,39)_ = 7.09, *p* = 0.011, η_p_^2^ = 0.15, π = 0.74) also originating from the higher sway irregularity of the older group in the two easiest dual-tasking conditions (DT_1_: +23% (old vs. young), *p* = 0.005; and DT_2_: +28% (old vs. young), *p* < 0.001). Figure 4 visualizes the descriptive statistics of the significant age effects. Complete descriptive statistics can be found in the Appendix A (“Age_effects.png”).

## 4. Discussion

Three hypotheses were tested in the current study. First, that an n-shaped relationship between cognitive demand and postural control is displayed in the PCA variables that quantify the irregularity and the tightness of the control of different movement components. Second, that in accordance with the minimal intervention principle the effects would emerge in specific task-relevant movement strategies. Third, that the efficacy of the control system of the older population is affected more by dual-tasking, due to the more limited cognitive resources. The results supported the first two hypotheses, but not the third. 

### 4.1. Dual-Tasking Effects

In our study both anteroposterior and mediolateral ankle strategies are the main contributors to the overall variance produced (around 78%), and therefore represent the main dynamics of the balancing task. The statistically significant result in ${\mathrm{SaEn}}_{1}^{PA}$ and the trends in the other three variables support H1 for the anteroposterior (AP) ankle sway (PM_1_). To our knowledge there is only one previous study comparing the same n-back DT conditions (n = 1, 2, 3) for tandem stances, which found comparable DT effects in the AP-COP displacement [10]. For the mediolateral (ML) ankle sway (PM_2_) both the irregularity of the control ${\mathrm{SaEn}}_{2}^{PA}$ and the timing variability of the interventions σ_2_ displayed statistical significance. In agreement with previous literature [10,12,14,15,16,20,53] and H1 the effects in ${\mathrm{SaEn}}_{2}^{PA}$ suggest higher automaticity [16,22,24,26] of the control system for medium difficulties followed by a decrease for harder dual-tasks. However, while ${\mathrm{SaEn}}_{2}^{PA}$ shows the expected n-shape trend, supporting H1, σ_2_ exhibits a steady increase with difficulty level. This steady increase of σ_2_ shows that the timing of the control interventions of the mediolateral ankle sway becomes more variable with increasing task difficulty, which would suggest a steady decrease in the tightness of the control. Due to the smaller base of support in mediolateral direction while tandem standing, it seems obvious that PM_2_ is one of the most critical movement strategies and is tightly controlled by an effective control system [40]. In contrast to the literature [10,12,14,15,16,20,53] and H1, this would suggest that although the automaticity displays the n-shape the efficacy of postural control is steadily reduced with increasing dual-task difficulty. The trends in ${\mathrm{SaEn}}_{2}^{PP}$ and N_2_ (n-shaped automaticity and steady decrease in the number of interventions) support this assumption; however, these results were not significant. 

Interestingly, while in our study all entropy measures display a peak in condition DT_2_, N*_k_* and σ*_k_* display their extrema in the DT_1_ condition. The AP-COP sway in the other study [10] also displayed the expected u-form with the least sway in the DT_2_ condition. This might suggest that in terms of sway and control irregularity, highest automatization is reached at the more difficult DT_2_ condition, whereas the tightest postural control (high N, low σ) is typically found in the DT_1_ condition while DT_2_ already shows signs of a decreased control tightness. It is important to note that this interpretation is speculative, since the peak values were only significantly different compared to extreme values, e.g., when comparing the irregularity of DT_2_ to the irregularity in ST or the most difficult DT_4_ condition; or when comparing the timing variability σ_2_ of ST to DT_4_ condition. Hence, the data does not allow us to make definitive statements when comparing non-extreme conditions and therefore the conditions that display peak-values could not be determined in a conclusive manner.

Furthermore, the absence of dual-tasking effects in ${\mathrm{SaEn}}_{3–6}^{PA}$ suggests that the irregularity of the neuromuscular control mechanisms of PM_1-2_, i.e., anteroposterior AP and mediolateral ML ankle sway, is affected in greater magnitude than the control irregularity of PM_3-6_. In addition, the ML ankle sway is the only movement component that shows significant DT effects in the timing variability of the neuromuscular system’s intervention. These results support H2, since PM_1_ and PM_2_ displayed the main dynamics and PM_2_ is probably the most task-relevant movement to be controlled. However, the absence of significant dual-tasking effects in ${\mathrm{SaEn}}_{1–4}^{PP}$ and N_1-4_ suggests that the sway regularity and the number of interventions are affected in greater magnitude in PM_5-6_, i.e., upper body rotation and hip/knee strategy, respectively. On the one hand, the minimal intervention principle suggests that the control system focuses on task relevant movements, thus we expect limitations in cognitive resources to affect these relevant components in greater magnitude. This would suggest that in terms of sway irregularity and the number of interventions of the control system, upper body rotations and the hip/knee strategy are very task-relevant. On the other hand, if PM_1-2_ are, as we assume, the most task relevant, these results would suggest that the control system manages to channel sufficient attention to PM_1-2_ to avoid significant changes in the sway irregularity and the number of control interventions. Thus, the cognitive resources available for higher order movement components would be limited resulting in the observed dual tasking effects in PM_5-6_.

### 4.2. Age-DT Interaction Effects

The two age-DT interactions in ${\mathrm{SaEn}}_{6}^{PP}$ and ${\mathrm{SaEn}}_{5}^{PA}$ show that in these two higher components the older population displayed the expected trend of increasing automatization followed by limitation due to cognitive resource competition [9,15,16]. However, the younger group only displayed subtle changes in the irregularity of sway and control, suggesting that they managed to divert a similar amount of attention to these movement components, while the older group experienced the n-shaped dual-tasking effects. In the other variables, both populations display similar dual-tasking effects.

### 4.3. Age Effects

In accordance with previous findings [40], the control system of the older population displayed a lower amount of control interventions for all conditions in mediolateral ankle sway (N_2_). However, only the age differences in the two easiest conditions (ST and DT_1_) were significant. Furthermore, the older participants only displayed minor decreases of N_2_ in this component that is critical for maintaining postural stability, while the younger participant’s number of control interventions decreased steadily. Hence, in contrast to H3, it is the younger age group that exhibits larger decreases in control interventions with increasing cognitive load. In addition, the trend in σ_2_ suggested that the younger participants exhibited a greater increase in timing variability when increasing cognitive load. Nevertheless, although the control tightness of the mediolateral ankle sway of the younger participants decreased more with increased cognitive load, they exhibited tighter postural control in all conditions in this critical component.

The age effects in ${\mathrm{SaEn}}_{3}^{PP}$ and ${\mathrm{SaEn}}_{6}^{PP}$ indicated that the younger population exhibited lower sway irregularity in the movement components that resemble an upper body retraction and a hip/knee-strategy, respectively. In the paradigm of movement automatization [16,22,24,26] this would mean that the postural control of PM_3/6_ was more automatized in the ST condition and then displayed the postulated increase in automaticity, followed by a decrease of sway irregularity due to resource competition. Here, the older participants followed the expected u-shaped pattern, while the younger participants seemed to maintain an unaltered focus of attention on trunk stability throughout the DT conditions. In accordance with previous findings [54,55], this could be interpreted as a sign of less variable and more effective control of trunk stability of the younger group.

### 4.4. Limitations

The neglected PM*_k_* (with *k* > 6) explains around 5% of the overall variance. Though it is possible that interesting aspects of the movement were neglected, we are confident that the main dynamics of the balancing task were captured. Furthermore, the PM*_k_* are linear movement components. Therefore, the interpretation of individual PM*_k_* must be done with caution, since they can only approximate the dynamics of real movements. 

## 5. Conclusions

The current study found varying non-linear relationships between postural control and increasing difficulty levels of cognitive demand, and different magnitudes of the relationships, as postulated by the minimal intervention principle. Hence, despite the overall n-shaped relationship the movement component most critical for postural stability displayed steady decreases in the tightness of the control. 

## Figures and Tables

**Figure 1 entropy-21-00070-f001:**
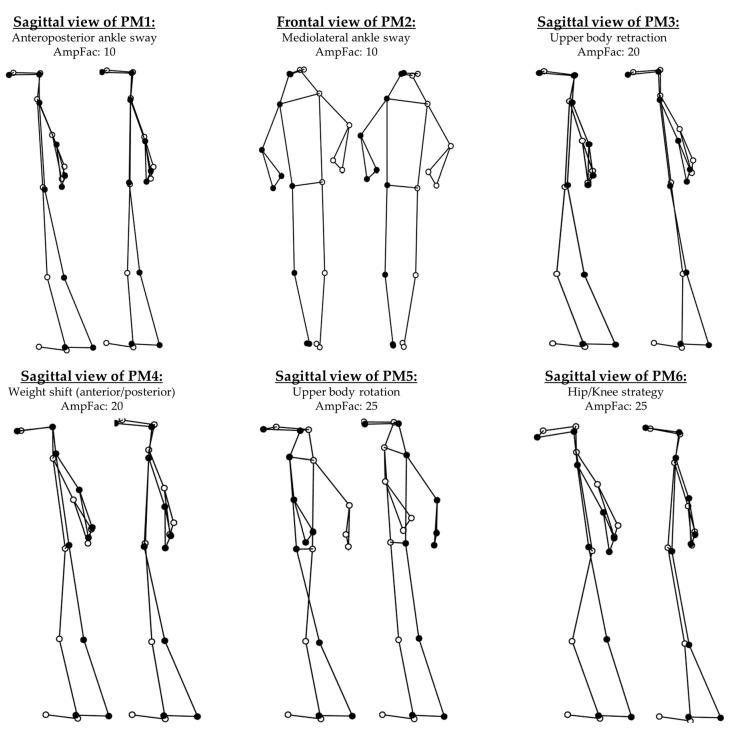
Visualization of the first six principal movements (PM_1_–PM_6_) of the tandem stance with respective amplification factors (AmpFac). For each PM the minimal and maximal deviation from the mean posture are displayed.

**Figure 2 entropy-21-00070-f002:**
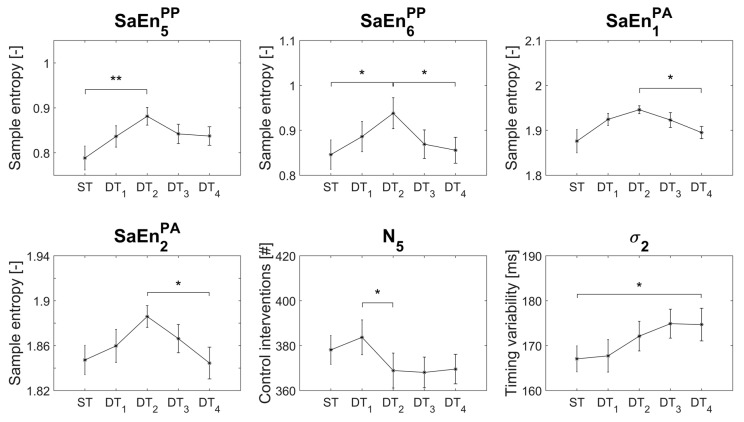
Post-hoc analysis and descriptive statistics of the variables that displayed dual-tasking effects. ${\mathrm{SaEn}}_{k}^{PP/PA}$ stands for sample entropy of the principal position PP or principal acceleration PA of the *k*th principal movement. N*_k_* and σ*_k_* stand for the number of control interventions and the timing variability of the interventions of the *k*th component, respectively. Significant post-hoc results are symbolized with asterisks. ST = single task; DT*_n_* = dual task with n-back auditory working task.

**Figure 3 entropy-21-00070-f003:**
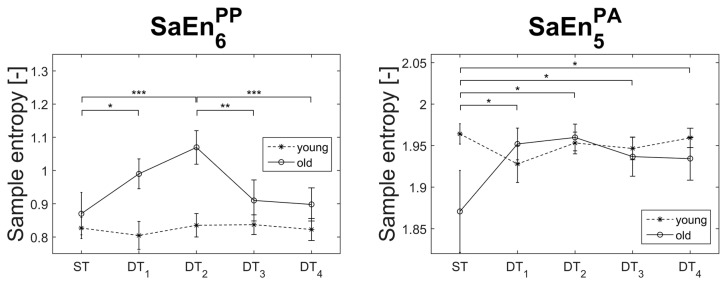
Descriptive statistics of the variables that displayed dual-tasking age interaction effects. ${\mathrm{SaEn}}_{k}^{PP/PA}$ stands for sample entropy of the principal position PP or principal acceleration PA of the *k*th principal movement. Significant post-hoc results are symbolized with asterisks (dual-tasking effects were found only in the older group). ST = single task; DT*_n_* = dual task with n-back auditory working task.

**Figure 4 entropy-21-00070-f004:**
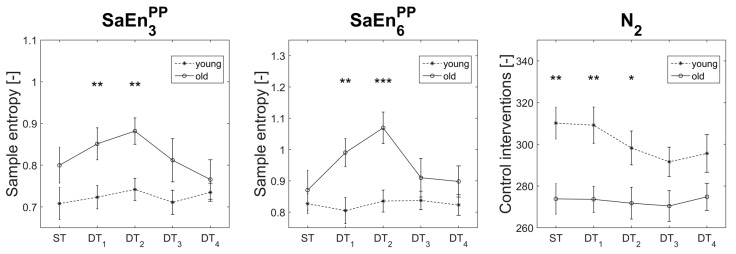
Descriptive statistics of the variables that displayed age effects. ${\mathrm{SaEn}}^{PP/PA}$ stands for sample entropy of the principal position PP or principal acceleration PA and PM*_k_* for the *k*th principal movement. N*_k_* stands for the number of control interventions in the *k*th component. Significant post-hoc results are symbolized with asterisks. ST = single task; DT*_n_* = dual task with n-back auditory working task.

**Table 1 entropy-21-00070-t001:** Description of the first six principal movements PM*_k_* that cumulatively describe over 95% of the overall variance. Significant effects in ${\mathrm{SaEn}}_{k}^{PP}$, ${\mathrm{SaEn}}_{k}^{PP}$, N*_k_* or σ*_k_* are symbolized by *PP*, *PA*, *N* or *σ,* respectively.

*k*	EV [%]	Effects	Main Strategy (Directions)	Specifications/Additional Features
1	51.1	*PA*	Ankle (anterior/posterior)	No visible motions in the rest of the body.
2	26.5	*PA*, *σ*	Ankle (medial/lateral)	No visible motions in the rest of the body.
3	9.7		Upper body (retraction)	Upper body leans back. Front knee (flexion/extension).
4	3.9		Weight shift (anterior/posterior)	Upper body shifted from over one foot to over the other.
5	2.6	*PP, N*	Upper body rotation	No visible motions in the rest of the body.
6	1.9	*PP, N*	Hip/Knee strategy	Flexion/extension in both hip and knee.

**Table 2 entropy-21-00070-t002:** Statistics describing dual-tasking effects of the variables ${\mathrm{SaEn}}_{k}^{PP},\;{\mathrm{SaEn}}_{k}^{PA}$, N*_k_*, and σ*_k_*. Significant effects are highlighted with bold font.

	${\mathrm{SaEn}}_{k}^{PP}$			${\mathrm{SaEn}}_{k}^{PA}$			N*_k_*			σ*_k_*		
	*p*	$\eta_{p}^{2}$	π	*p*	$\eta_{p}^{2}$	π	*p*	$\eta_{p}^{2}$	π	*p*	$\eta_{p}^{2}$	π
**PM** _1_	0.062 ^1^	0.06	0.64	**0.016** *^,2^	**0.09**	**0.76**	0.147 ^1^	0.04	0.50	0.190 ^2^	0.04	0.37
**PM** _2_	0.228	0.04	0.45	**0.028** *	**0.07**	**0.76**	0.082	0.05	0.62	**0.024** *	**0.07**	**0.77**
**PM** _3_	0.054	0.06	0.68	0.101 ^2^	0.05	0.53	0.363 ^1^	0.03	0.33	0.403	0.03	0.03
**PM** _4_	0.128 ^1^	0.05	0.53	0.082	0.05	0.62	0.147	0.04	0.52	0.116	0.05	0.56
**PM** _5_	**0.003** **	**0.10**	**0.92**	0.303 ^1^	0.03	0.34	**0.011** *^,1^	**0.09**	**0.83**	0.216 ^1^	0.04	0.42
**PM** _6_	**0.017** *	**0.07**	**0.80**	0.099 ^2^	0.05	0.52	**0.030** *	**0.07**	**0.75**	0.065 ^1^	0.06	0.64

^1^ Huynh–Feldt corrected; ^2^ Greenhouse–Geisser corrected. * *p* < 0.05. ** *p* < 0.01.

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
