# Peer review of "The Effect of Cognitive Resource Competition Due to Dual-Tasking on the Irregularity and Control of Postural Movement Components"

_entropy, 2019, doi:10.3390/e21010070_

Round 1

Reviewer 1 Report

some questions

How were evaluations performed, barefoot or footwear?

The base area of support, was taken into account?

 suggestion:

many references too old, should replace if possible.

Author Response

1. How were evaluations performed, barefoot or footwear?

The participants stood barefoot. See changes in lines 139-142.

2. The base area of support, was taken into account?

In this study, the base of support depends on the size of the feet of the participants. We normalized the data of the participants to their respective height (see 3.1.1. Pre-processing) which is to some extent related to the size of the feet. However, future studies using PCA might indeed benefit from specific “base-of-support-normalizations”, which could be achieved via specific measurements.

3. many references too old, should replace if possible:

We agree that many references are older. However, many of them can be considered standard references with relevant conclusions in their respective field and are still cited often. Nevertheless, we exchanged most citations prior to the year 2000 with newer literature (see citations 1-4, lines 33-34) and expanded some of the other references (line 38 and line 49).

Reviewer 2 Report

The manuscript entitled “the effect of cognitive resources competition due to dual-tasking on the irregularity and control of postural movement components” is interesting for entropy readers and well written. In my opinion this paper deserves to be published but some of my comments can help to improve it slightly.

1.     Please consider to include the following references to strengthen the introduction.

Estevan, I., Gandia, S., Villarrasa-Sapiña, I., Bermejo, J. L., & García-Massó, X. (2018). Working Memory Task Influence in Postural Stability and Cognitive Function in Adolescents. Motor control22, 425-435.

Bustillo-Casero, P., Villarrasa-Sapiña, I., & García-Massó, X. (2017). Effects of dual task difficulty in motor and cognitive performance: Differences between adults and adolescents. Human movement science55, 8-17.

Lacour, M., Bernard-Demanze, L., & Dumitrescu, M. (2008). Posture control, aging, and attention resources: models and posture-analysis methods. Neurophysiologie Clinique/Clinical Neurophysiology38(6), 411-421.

2.     Line 139-142: I do not understand which foot was placed rear and which on the front. Dominant foot rear or in the front? Please explain it better.

3.     Line 142-145: please explain with more detail how the participants carried out the n-back working memory task. I think that it could be nice to include the number of sequences that the subjects performed in each condition (mean and SD).

4.     Line 152: please include city and country for SPSS software.

5.     Line 155: “…difficulties and fatigue effects” please add a reference.

6.     Line 159: “…(=84 columns)” is it the results of 24 markers * 3 dimensions? Please explain it.

7.     Lines 211-213: all the dependent variables pass normality and homocedasticity assumptions?

8.     Figure 1: could be nice to include the % of explained variance of each PM.

9.     If no limits of figures and tables are established by the journal, I suggest to add a new figure with pairwise comparisons of the interaction effect between DT and age.

Author Response

1.     Please consider to include the following references to strengthen the introduction.

Thank you for the comment. We included Bustillo et. at and Lacour et al. (see lines 38 and 49). However, we did not include Estevan et al. because the participants in the study are adolescents aged 13-16, which does not correspond to any of our age groups.

2.     Line 139-142: I do not understand which foot was placed rear and which on the front. Dominant foot rear or in the front? Please explain it better.

The participants tried out both tandem stance positions before the measurements (left foot in front and right foot in front) and were instructed to stand in the position that felt more comfortable. We clarified this in lines 139-142.

3.     Line 142-145: please explain with more detail how the participants carried out the n-back working memory task. I think that it could be nice to include the number of sequences that the subjects performed in each condition (mean and SD).

Thank you for the comment. In each trial the participants listened to one sequence of 26 numbers, with 6 randomized n-back stimuli. We included this information in the changes in lines 143-149.

4.     Line 152: please include city and country for SPSS software.

Thank you for the comment. Changes were made in lines 155-156.

5.     Line 155: “…difficulties and fatigue effects” please add a reference.

We added one reference for each (starting difficulties and end effects). The word “fatigue” could be misleading, as it can be of various forms, muscular, cognitive, neurological, etc. To avoid confusion, we changed “fatigue” to “teleoanticipation effects due to the end of the trial” (see lines 159-160).

6.     Line 159: “…(=84 columns)” is it the results of 24 markers * 3 dimensions? Please explain it.

We added the information in line 164.

7.     Lines 211-213: all the dependent variables pass normality and homocedasticity assumptions?

The criteria for homoscedasticity are fulfilled for all variables. Normality assumptions are fulfilled for almost all variables and trials. Specifically, some PA show some exceptions, however only few (e.g. one out of five conditions). Since ANOVAS are rather robust to violations of the normality assumption we considered repeated measures ANOVA to be suitable for all variables.

8.     Figure 1: could be nice to include the % of explained variance of each PM.

Thank you for the comment. We tried to include the % in Figure 1. However, we found that the figure lost clarity and we would be repeating information contained in Table 1. Therefore, we did not make the change and hope for your understanding.

9.     If no limits of figures and tables are established by the journal, I suggest to add a new figure with pairwise comparisons of the interaction effect between DT and age.

Thank you for the suggestion. We added two figures. One containing the post hoc analysis of the significant interaction effects (Figure 3, line 273) and one containing the post hoc analysis of the significant age effects (Figure 4, line 291).

Round 2

Reviewer 2 Report

The authors have taken into account and addressed all my (relevant) concerns. I would like to congratulate the authors on their work.